# Determinants of feto-maternal outcomes of antepartum hemorrhage among women who gave birth in Awi zone public hospitals, Ethiopia. A case-control study

**Ambaye Minayehu Zegeye**[1]*, **Yibelu Bazezew**[2], **Ashete Adare**[3], **Paulos Jaleta**[4], **Wale Kumlachew**[4], **Seid Wodajo Liben**[1], **Yaregal Dessalew Tarik**[1], **Getahun Degualeh Kebede**[1], **Yilkal Dagnaw**[1], **Fentahun Tamene Zeleke**[5], **Dawit Misganaw Belay**[1]

**1** Department of Midwifery, College of Health Sciences, Assosa University, Assosa, Ethiopia, **2** Department of Midwifery, College of Health Sciences, Debre Markos University, Debre Markos, Ethiopia, **3** Department of Anatomy, College of Biomedical Sciences, Arsi University, Asella, Ethiopia, **4** Department of Nursing, College of Health Sciences, Assosa University, Assosa, Ethiopia, **5** Department of Midwifery, College of Health Sciences, Wolkite University, Wolkite, Ethiopia

* abelminayehu15@gmail.com

## Abstract

### Background

Antepartum hemorrhage continues to be a major cause of maternal and perinatal morbidity and mortality in developing countries including Ethiopia and it complicates 2–5% of all pregnancies with an increased rate of maternal and perinatal morbidity and even mortality. Despite many activities, still, poor fetomaternal outcomes of antepartum hemorrhage are still there. Moreover, studies around the current study area emphasize the magnitude and associated factors for antepartum hemorrhage rather than its feto-maternal outcomes. Thus, there is a need to identify the determinants associated with the fetomaternal outcomes of antepartum hemorrhage to guide midwives and obstetricians in the early diagnosis and treatment.

### Method

An institution-based case-control study was conducted in four-year delivery charts diagnosed with antepartum hemorrhage from April 2, 2022, to May 12, 2022, at Awi Zone public hospitals. To see the association between dependent and independent variables logistic regression model along with a 95% confidence interval (CI) and a p-value of <0.05 were used.

### Result

No antenatal care follow-up (AOR: 2.5, 95% CI 1.49–4.2), rural residence (AOR: 1.706, 95%CI 1.09–2.66), delay to seek care >12 hours (AOR: 2.57, 95% CI: 1.57–4.23) and advanced maternal age (AOR: 3.43, 95% CI 1.784–6.59) were significant factors associated with feto-maternal outcomes of antepartum Hemorrhage.

**Data Availability Statement:** All relevant data are within the paper and its Supporting Information files.

**Funding:** We received a small grant from Assosa University and Debre Markos University for data collection, these institutions have no role in study design, data collection, and analysis, decision to publish, or preparation of the manuscript.

**Competing interests:** The authors have declared that no competing interests exist.

## Conclusion

This study revealed that rural residence, delay in seeking the care of more than 12 hours, not having antenatal care follow up and advanced maternal age were significant factors associated with feto-maternal outcomes of Antepartum hemorrhage.

## Recommendation

The findings of our study suggest the need for health education about the importance of antenatal care follow-up which is the ideal entry point for health promotion and early detection of complications, especially for rural residents.

## Introduction

Antepartum hemorrhage is bleeding through the birth canal from the 28th week of gestation to the delivery of the baby. Mainly placenta previa and abruption-placenta cause antepartum hemorrhage (APH) [1], but in a small proportion, local lesions of the cervix and vagina, rupture of vasa previa, and uterine rupture might be the causes it [2, 3]. Antepartum hemorrhage complicates 2–5% of all pregnancies with increased rates of feto-maternal morbidity and mortality [2]. Placenta previa is responsible for one-third of APH with the incidence ranging from 0.5 to 1% [4]. A recent study in Addis Ababa, Ethiopia showed that APH affects 3.7% of all deliveries [5].

APH has been associated with several maternal and fetal complications that include post-partum hemorrhage (PPH), hemorrhagic shock, anemia, sepsis, premature labor, prematurity, stillbirth, fetal hypoxia, low birth weight, and neonatal intensive care unit admission [1]. Moreover, APH continues to be one of the leading causes of maternal death, accounting for 50% of the reported 500,000 maternal deaths worldwide per year [6–8]. It also accounts for 30% of the direct causes of maternal death [9–11].

From 2000 to 2017, the global maternal mortality rate (MMR) declined by 38%, from 451,000 to 295,000. Despite the substantial decrease, MMR remains relatively high in low- and middle-income nations. Sub-Saharan African nations are responsible for almost 60% of the predicted global maternal mortality in 2017 [12].

The absolute number of maternal deaths in Ethiopia has decreased by 55%, from 31,000 in 1990 to 14,000 in 2017 [3, 12]. In the previous two decades, the country has made significant progress in reducing maternal mortality. The MMR has decreased from roughly 871 deaths in 2000 to 401 deaths in 2017; nonetheless, there is a significant difference within regional states [13–15]. Because of these geographical variances, the country has failed to meet one of the MDG's aims of lowering MMR to 267 by 2015 [3].

Direct obstetric causes such as bleeding, infection, unsafe abortion, hypertensive disorders of pregnancy, and obstructed labor account for almost three-quarters of maternal deaths in low-income countries [16]. Studies showed that advanced maternal age, parity, previous history of APH, rural residence, no antenatal care (ANC), delay seeking care, and previous cesarean delivery are risk factors for the occurrence of APH [7, 17, 18].

Ethiopia launched the maternal death surveillance and response system (MDSR) in 2013 to provide real-time data on the patterns and trends of preventable maternal death and achieve the sustainable development goal target 3.1 (a lower maternal mortality ratio of less than 70 per 100 000 live births by 2030) [19–21]. MDSR is a continual process of recognizing maternal

mortality, acquiring information on the causes and determinants of those deaths, and analyzing the data to prevent similar deaths in the future [22].

In tandem with the establishment of the MDSR, Ethiopia has initiated several measures to reduce unnecessary maternal deaths, such as the construction of maternity waiting rooms within health facilities [23, 24], as well as the availability of free transportation and maternity services [25]. Despite all of these efforts and measures, MMR in Ethiopia remains unacceptably high [26].

Even if several studies were investigated in the study area, all emphasized the magnitude and its associated factors for APH. However, none of the studies incorporates the feto-maternal outcomes of APH. Therefore, this study aimed to identify the determinants of feto-maternal outcomes of APH among women who were complicated by antepartum hemorrhage in Awi Zone public hospitals.

## Material and methods

### Study area and period

This study was conducted at Awi Zone public hospitals, namely, Injibara General Hospital, Chagni Primary Hospital, Dangila Primary Hospital, Agew Gimjabet Primary Hospital, and Jawe Primary Hospital, from April 2–May 12, 2022. Awi Zone is one of the administrative zones in the Amhara regional state, with the capital city of Injibara. It is located in the northwestern part of Ethiopia, 447 kilometers away from Addis Ababa, the capital city of Ethiopia. According to the Central Statistical Agency of Ethiopia, in 2007, more than one million people populated the Awi Zone.

### Study design

An institution-based case-control study was conducted at Awi Zone public hospitals.

### Source and study population

The source population of this study was medical records of mothers who gave birth in Awi Zone public hospitals, while the study population was mothers' medical records diagnosed with APH from March 2018 through 2022 in Awi Zone public hospitals.

### Sample size determination

Epi-Info version 7.2.1 was used to determine the sample size, considering the following assumptions: 95% confidence interval, power of 80%, an adjusted odds ratio of 3.7, the ratio of cases to control is 1:1, and taking 2.8% of the proportion of APH associated with stillbirth (one of the feto-maternal outcomes of APH) from a case-control study conducted in Nepal [27]. The final calculated sample size was 448 medical record charts (224 cases and 224 controls).

### Sampling technique and procedure

A complete ascertainment of cases and a simple random sampling technique were employed for the selection of controls. The four-year (March 1, 2018–March 30, 2022) medical record numbers of all mothers who had APH were traced from the hospitals' delivery logbook registry. After reviewing their completeness, all medical record numbers were grouped into cases and controls. All the cases were ascertained completely, and controls were entered into Microsoft Excel to apply a computer-generated simple random sampling technique.

Six hundred fifty-one APH cases were reported in the health management information systems of the five hospitals. Then proportional allocation was employed to get the required sample size in each hospital, as shown in the following figure (Fig 1).

## Inclusion and exclusion criteria

All women's medical charts with a diagnosis of APH were included, while medical charts with missed information, multiple pregnancies, and confirmed intrauterine fetal death before the onset of APH were excluded.

## Study variables

**Dependent variables.** Feto-maternal outcomes of antepartum hemorrhage (good or poor).

**Independent variable.** Socio-demographic factors.

- Age
- Residence

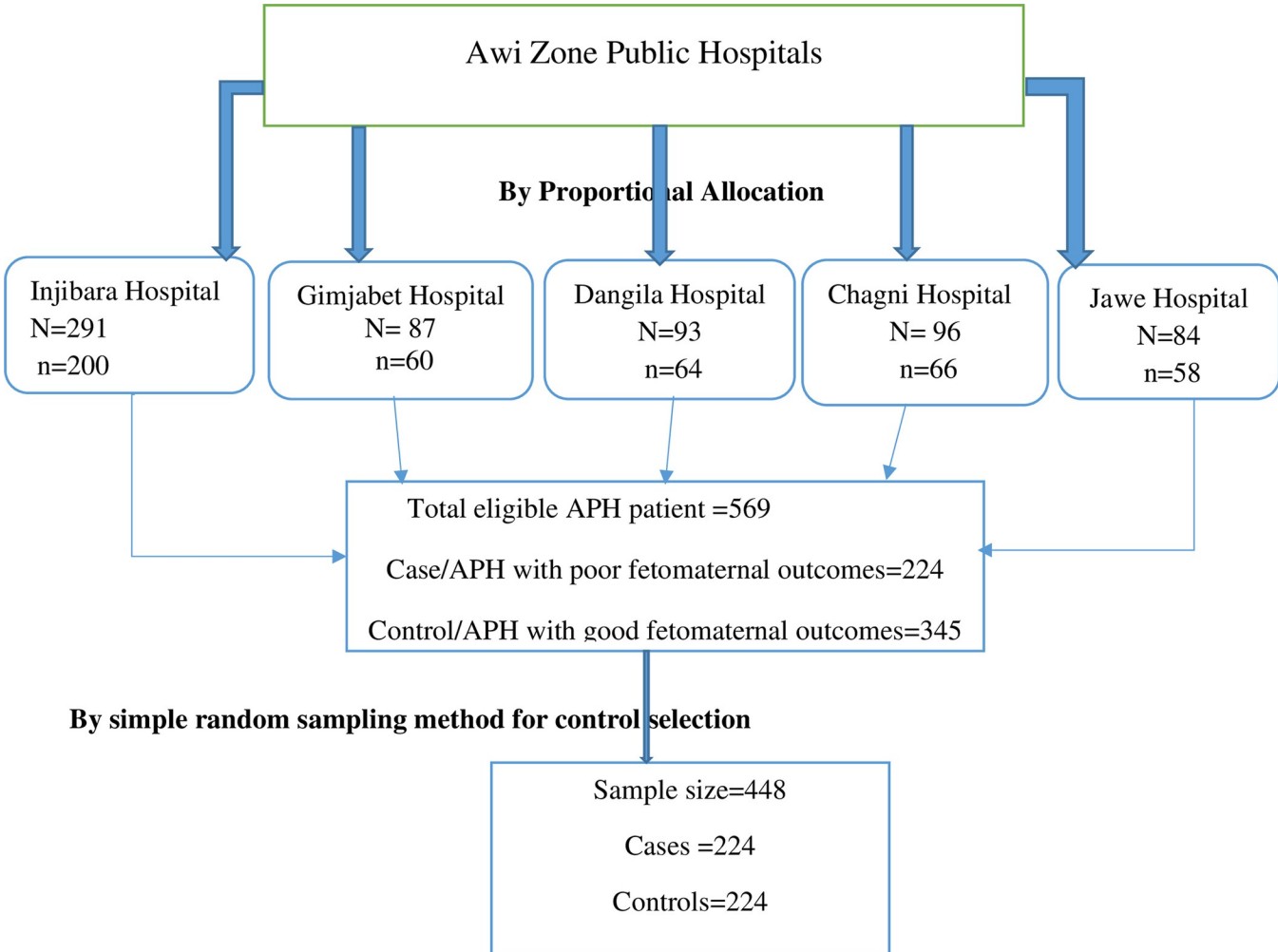

**Fig 1. Schematic presentation of the sampling procedure for the selection of study subjects at Awi zone hospitals, North West, Ethiopia, 2021.**

- ■ Education

- ■ Occupation

- ■ Marital status

Health system-related factors

- ■ Referral system

- ■ Distance traveled

- ■ Blood transfusion service

- ■ Neonatal intensive care unit service

- ■ Prolonged hospital stay

- ■ ANC follow-up

Clinical factors

- ■ History of hypertension disorder during pregnancy

- ■ Previous history of cesarean delivery

- ■ Mode of delivery

- ■ History of curettage

- ■ History of APH

- ■ Types of current APH, anemia, and hysterectomy

- ■ History of abdominal trauma

- ■ Parity and gravidity

- ■ Gestational age

- ■ PPH, coagulopathy

- ■ Number of pregnancies

## Operational definitions

**Poor feto-maternal outcome:** this is a medical condition in which either the mother or the newborn suffers from at least one of the complications due to APH. The maternal complications include death, hemorrhagic shock, PPH, anemia, the need for a blood transfusion, a long hospital stay, and peripartum hysterectomy. Whereas neonatal complications might be preterm birth, low birth weight (less than 2500 gm), meconium aspiration syndrome, low APGAR score (less than 7), admission to the NICU, and newborn death (stillbirth) [17].

**Antepartum hemorrhage:** APH is defined as bleeding from or into the genital tract after the 24th week of pregnancy but before the delivery of the baby (last baby in case of multiple pregnancies) [28].

**Abruptio placenta:** an obstetric complication characterized by the detachment of a normally implanted placenta before delivery of the fetus [29].

**Placenta previa:** an obstetric complication characterized by placental implantation into the lower segment of the uterine wall, covering part of or the entire cervix [29].

**Prolonged hospital stay:** admission of the mother for more than one day for spontaneous vaginal delivery and more than seven days for complicated delivery [17].

**Cases:** all women with the antepartum hemorrhage who developed poor feto-maternal outcomes.

**Control:** all women with the antepartum hemorrhage who had good feto-maternal outcomes.

## Data collection tools

A structured, validated, and pretested questionnaire adapted from related works of literature [6, 17, 30, 31] was used to collect data. Selected socio-demographic factors, health system-related factors, clinical factors, and feto-maternal outcomes of APH-related information were extracted from the medical record.

## Data quality management

To ensure the quality of the data, one-day training about the objective of the study and proper handling of the data was given to data collectors and supervisors. A pretest among 5% of the total sample size was conducted at Injibara General Hospital one week before the data collection period. These were the women's medical records of antepartum hemorrhage before March 2017, which were not included in the actual data collection. The questionnaire was modified based on the pre-test, and variables like the grade of abruption of the placenta, level of income, and organ function test were excluded from the questionnaire. Supervision was conducted throughout data collection, and the data was crosschecked for completeness by the data collectors, supervisors, and principal investigator.

## Data analysis and processing

The data was coded and entered into Epi Data version 4.6, then exported to SPSS version 25.0 for analysis. Descriptive statistics like frequencies and percentages were presented with texts, tables, and simple bar graphs. A bivariable logistic regression analysis was performed to see the association between each independent variable and the outcome variable. Multi-collinearity was checked by using the variance-inflation factor (VIF), which ranges from 1.05 to 1.28. Model fitness was checked using the Hosmer-Lemeshow goodness of fit (P > 0.05). An adjusted odds ratio along with a 95% confidence interval assessed the degree of association between independent and dependent variables. Those variables with a p-value of less than 0.05 were considered statistically significant.

## Ethical clearance

Ethical clearance was obtained from the ethical review committee of Debre Markos University College of Health Sciences. Written permission to conduct the study was obtained from subsequent hospital administrators after explaining the purpose and ethical process of the study. Moreover, there are no invasive procedures in this study, and it was conducted following the Declaration of Helsinki.

## Results

### Sociodemographic characteristics of cases and controls

There were 25,152 deliveries in the four years at Awi Zone public hospitals. Among these, 651 had antepartum hemorrhage which accounts for 2.6% of the total deliveries. The overall mean age for our sample (448) was 30.89±5.836 years, and women with cases (poor feto-maternal outcomes) had a significantly higher mean age (32.28, SD = 5.77 years) than their unaffected counterparts (29.49, SD = 5.77 years). Among cases, 50.3% of the women were in the age range

**Table 1. Sociodemographic characteristics of the cases and controls in Awi Zone public hospitals, 2022 (N = 448).**

| Variables | Category | Cases (n = 224) | Controls (n = 224) |
|---|---|---|---|
| | | Frequency (%) | Frequency (%) |
| Age | 18–24 | 33(31.7%) | 71(68.3%) |
| | 25–29 | 32(48.5%) | 34(51.5%) |
| | 30–35 | 87(50.3%) | 86(49.7%) |
| | >35 | 72(64.7%) | 33(31,4%) |
| Residence | Urban | 79(35.3%) | 131(58.5%) |
| | Rural | 145(64.7%) | 93(41.5%) |
| Educational level | Can't read &write | 166(62.4%) | 100(37.6%) |
| | Attended grades 1–8 | 22(28.6%) | 55(71.4%) |
| | Attended grades 9–12 | 17(29.8%) | 40(70.2%) |
| | Diploma & above | 19(39.6%) | 29(60.4%) |
| Occupation | Housewife | 140(62.5%) | 113(50.4%) |
| | Governmental employee | 14(6.3%) | 30(13.4%) |
| | Student | 8(3.6%) | 13(5.8%) |
| | Merchant | 27(12.1%) | 46(20.5%) |
| | Private employee | 30(13.4%) | 11(4.9%) |
| | Daily laborer | 5(2.2%) | 11(4.9%) |
| Marital status | Married | 214(95.6%) | 210(93.7%) |
| | Single | 5(2.2%) | 8(3.6%) |
| | Divorced | 5(2.2%) | 6(2.7%) |

of 30–35, and 64.7% were rural residents. Out of the cases, regarding marital status, occupation, and education status, 95.6%, 62.5%, and 62.4% were married, housewives, and unable to read and write, respectively (Table 1).

## Clinically related characteristics of the cases and controls

Concerning parity, 117 (52.2%) and 16 (7.2%) of cases were grand multipara and primipara respectively, whereas 59 (26.3%) and 50 (22.3%) of controls were grand multipara and primipara respectively. The most common causes of APH were placenta previa (PP) (52.2%), abruption placenta (AP) (42.2%), and uterine rupture (5.4%). About seven (3.1%), nine (4%), five (2.2%), and six (3.1%) of cases had a history of previous APH, history of dilatation and curettage (D&C), history of hypertensive disorder of pregnancy (HDP), and history of previous cesarean delivery (C/D) respectively. With no significant difference 5 (2.2%), 8 (3.6%), 10 (4.5%), and 8 (3.6%) controls had a history of previous APH, history of dilatation and curettage, history of hypertensive disorder of pregnancy, and history of previous cesarean delivery respectively. One hundred thirteen (50.4%) newborns delivered in mothers with cases were preterm (Table 2).

## Health system-related characteristics of the cases and controls

About 124 (55.4%) cases and 190 (84.8%) controls had their regular ANC follow-up. From these, 56 (45.2%) cases and 132 (69.5%) controls had at least four ANC visits. For 169 (75.4%) cases and 122 (54.4%) controls, their mode of delivery was a caesarian section. Thirty-three (14.7%) cases and 46 (20.5%) controls with gestational age less than 37 weeks had been administered dexamethasone for lung maturity (Table 3).

**Table 2. Clinical-related characteristics of cases and controls in Awi Zone public hospitals, 2022 (N = 448).**

| Variable | Category | Cases n = 224 | Controls n = 224 |
|---|---|---|---|
| | | Frequency (%) | Frequency (%) |
| Parity | 1 | 16 (7.2%) | 50(22.3%) |
| | 2–4 | 91(40.6%) | 115(51.4%) |
| | 5 and above | 117(52.2%) | 59(26.3%) |
| Gestational age | <37 week | 113(50.4%) | 0(0%) |
| | ≥ 37 week | 111(49.6%) | 224(100%) |
| Cause of a current APH | PP | 110(49.1%) | 124(55.4%) |
| | AP | 90(40.2%) | 100(44.6%) |
| | Uterine rupture | 24(10.7%) | 0(0%) |
| History of previous APH | Yes | 7(3.1%) | 5(2.2%) |
| | No | 217(96.9%) | 219(97.8%) |
| History of D&C | Yes | 9(4%) | 8(3.6%) |
| | No | 215(96%) | 216(96.4%) |
| History of HDP | Yes | 5(2.2%) | 10(4.5%) |
| | No | 219(97.8%) | 214(95.5%) |
| History of a previous C/D | Yes | 6(2.7%) | 8(3.6%) |
| | No | 218(97.3%) | 216 (96.4%) |

## Maternal outcomes of antepartum hemorrhage

Of the total 224 cases, 31 (13.8%), 73 (32.4%), and 48 (21.4%) of them had developed PPH, hemorrhagic shock, and postpartum severe anemia, respectively. In these cases, 17 (7.6%) hysterectomy surgeries were performed, and 2 (0.9%) maternal deaths were reported. Fifty-four (24.1%) cases had been transfused with blood and 35 (15.6%) women had been admitted more than 7 days due to complications related to APH (Fig 2).

**Table 3. Health system-related factors of the cases and controls in Awi Zone public hospitals, 2022.** (N = 448).

| Variables | Category | Cases n = 224 | Controls n = 224 | Total N = 448 |
|---|---|---|---|---|
| | | Frequency (%) | Frequency (%) | Frequency (%) |
| ANC | No | 100(44.6%) | 34(15.2%) | 134(29.9%) |
| | Yes | 124(55.4%) | 190(84.8%) | 314(70.1%) |
| Number of ANC visits | I | 14(11.2%) | 5(2.6%) | 19(6.1%) |
| | II | 27(21.8%) | 17(8.9%) | 44(14%) |
| | III | 27(21.8%) | 36(18.9%) | 63(20%) |
| | IV& above | 56(45.2%) | 132(69.6%) | 188(59.9%) |
| Was remember LNMP | No | 120(53.6%) | 53(23.7%) | 173(38.6%) |
| | Yes | 104(46.4%) | 171(76.3%) | 275(61.4%) |
| Mode of delivery | C/D | 169(75.4%) | 122(54.4%) | 291(65%) |
| | SVD | 48(21.4%) | 96(42.9%) | 144(32.1%) |
| | Instrumental | 7(3.1%) | 6(2.7%) | 13(2.9%) |
| Referral from another health facility | No | 66(29.5%) | 110(49.1%) | 176(39.3%) |
| | Yes | 158(70.5%) | 114(50.9%) | 272(60.7%) |
| Iron folate supplementation | Yes | 119(53.1%) | 190(84.8%) | 309(68.97%) |
| | No | 105(46.9%) | 34(14.2%) | 139(31.03%) |
| Dexamethasone given | Yes | 33(14.7%) | 46(20.5%) | 79(17.6%) |
| | No | 191(85.3%) | 178(79.5%) | 369(82.4%) |

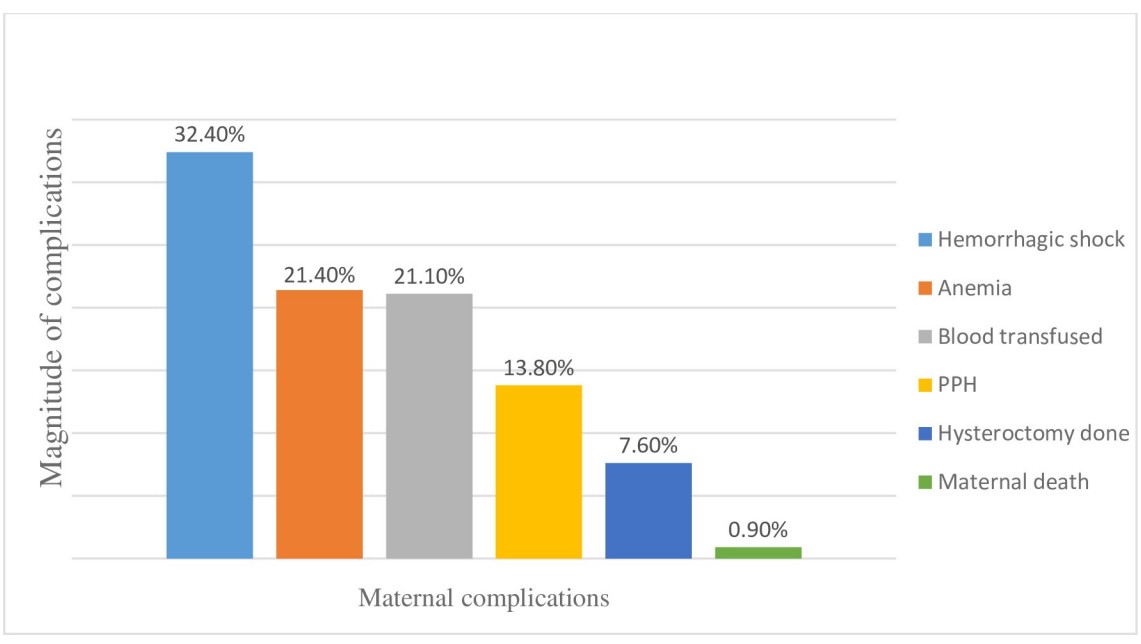

**Fig 2. Maternal complications of antepartum hemorrhage among 224 cases at Awi Zone public hospitals, 2022.**

### Fetal outcomes of antepartum hemorrhage

Two hundred thirty-one (51.6%) newborns were males. About 78 (17.4%) were stillborn. The first and fifth-minute APGAR scores were less than seven for 143 (63.8%) and 86 (38.1%) newborns, respectively. Regarding birth weight, 97 (43.3%) newborns were less than 2500 grams. Out of the live newborn cases, 80 (35.7%) were transferred to the NICU (Table 4).

### Fetal complications of antepartum hemorrhage

Regarding fetal complications, among cases 32% developed birth asphyxia, 25% were delivered prematurely, 22% were delivered with low birth weight, and 17% were stillborn (Fig 3).

**Table 4. Fetal outcomes of APH among women who gave birth in Awi Zone public hospitals, Ethiopia 2022, (n = 448).**

| Fetal outcome | Category | Cases n = 224 Frequency (%) | Controls n = 224 Frequency (%) | Total N = 448 Frequency (%) |
|---|---|---|---|---|
| Birth outcome | Alive | 146 (65.2%) | 224 (100% | 370 (82.6%) |
| | Stillbirth | 78 (34.8%) | 0 (0%) | 78 (17.4%) |
| Sex | Female | 109 (49.7%) | 108 (48.2%) | 217 (48.4%) |
| | Male | 115(51.3%) | 116 (51.8%) | 231 (51.6%) |
| Birth weight | 1000–1499 | 4 (1.8%) | 0 (0%) | 4(0.9%) |
| | 1500–2499 | 93(41.5%) | 0 (0%) | 93(20.8%) |
| | ≥2500 | 127 (56.7%) | 224 (100%) | 351 (78.3%) |
| APGAR score at 1$^{st}$min | <7 | 143(63.8%) | 0 (0%) | 143 (31.9%) |
| | >7 | 81(36.2%) | 224(100%) | 305 (68.1%) |
| APGAR score at 5$^{th}$min | <7 | 138 (61.9%0 | 0 (0%) | 138 (30.9%) |
| | >7 | 86 (38.1%) | 224(100%) | 309 (69.1%) |
| NICU admission | Yes | 80 (35.7%) | 0 (0%) | 80 (17.9%) |
| | No | 144 (64.3%) | 224 (100%) | 368 (82.1%) |

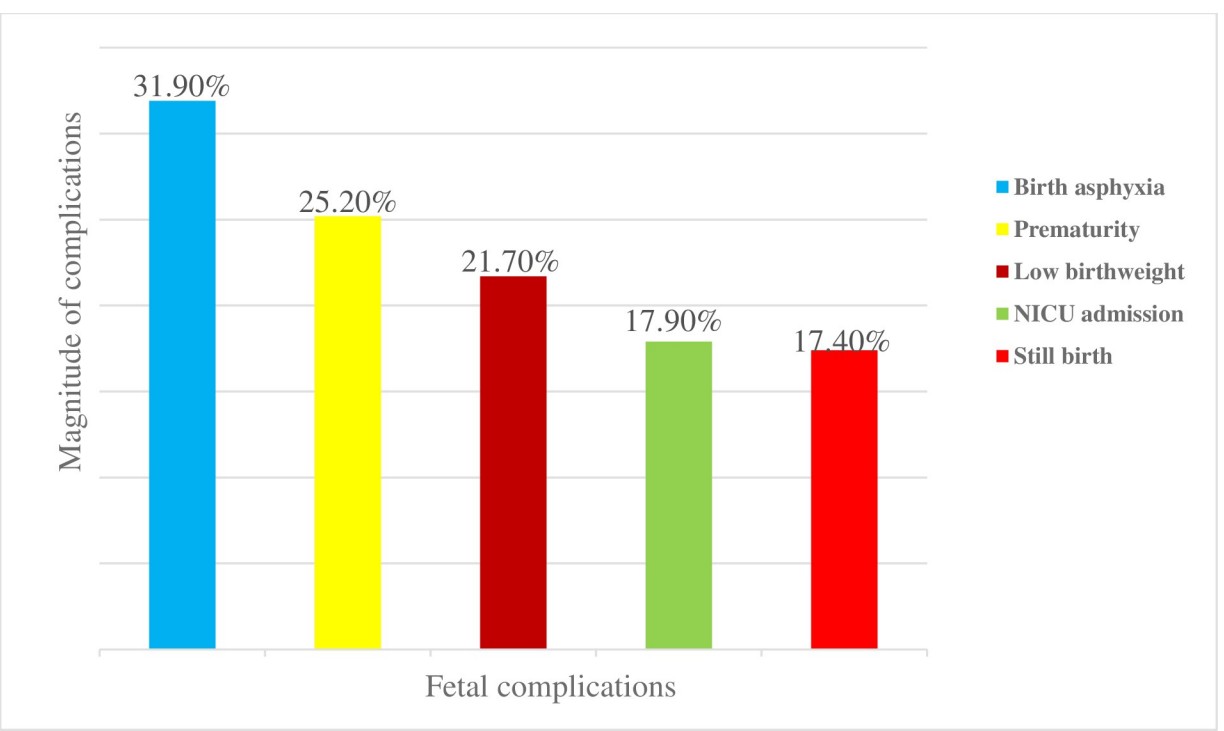

**Fig 3. Fetal complications of APH among women who gave birth in Awi Zone public hospitals, Ethiopia 2022.**

### Determinants of feto-maternal outcomes of APH

In the bivariable analysis, maternal age (being between 30–34 years and greater than 35 years), residence, mode of delivery, previous history of hypertension, educational level of the mother, dexamethasone administration, referral from another health facility, gravidity, delay in seeking care for more than 12 hours, and not having ANC follow-up was significantly associated with poor feto-maternal outcomes of APH. However, in the multivariable analysis, delaying seeking care for more than 12 hours, not having ANC follow-up, advanced maternal age (> 35 years), and rural residence remained significantly associated with poor feto-maternal outcomes of APH.

The odds of poor feto-maternal outcomes among mothers who delayed seeking care for more than 12 hours were 2.57 times higher as compared to their counterparts (AOR = 2.57, 95% CI: 1.57, 4.23). The odds of poor feto-maternal outcomes among mothers who had no ANC follow-up were about 2.5 times higher than those of women who had ANC follow-up (AOR = 2.5, 95% CI: 1.49, 4.2). Women with advanced maternal age (>35) were 3.429 times more likely to develop poor feto-maternal outcomes compared to women with an age group less than 35 (AOR = 3.43, 95% CI: 1.79, 6.59). Women residing in rural areas were 1.7 times more likely to develop poor feto-maternal outcomes as compared to women residing in urban areas (AOR = 1.7, 95% CI 1.09, 2.66). However, after adjusting the confounders, mode of delivery, previous history of hypertension, educational level of the mother, dexamethasone administration, referrals from other health facilities, and gravidity were not significantly associated with poor feto-maternal outcomes of APH (Table 5).

### Discussion

This study investigated the association between determinants and feto-maternal outcomes of APH and revealed that rural residence, delay seeking care >12 hours, not having ANC

**Table 5. Bivariable and multivariable logistic regression analysis of determinants of feto-maternal outcomes of APH among women who gave birth in Awi Zone public hospitals, Ethiopia, 2022, (N = 448).**

| Variable | Category | Fetomaternal outcomes | | 95% CI | | P value |
|---|---|---|---|---|---|---|
| | | Poor | Good | COR | AOR | |
| Maternal age | >35 | 72 | 33 | 4.69 (2.5,8.0) | 3.43 (1.78, 6.59) | 0.000* |
| | 30–35 | 87 | 86 | 2.08 (1.3,3.4) | 1.76 (1.00, 3.09) | 0.051 |
| | 25–29 | 32 | 34 | 1.9 (0.28,3.4) | 1.78 (0.88, 3.59) | 0.108 |
| | <25 | 34 | 70 | 1 | | |
| Residence | Rural | 146 | 92 | 2.7 (1.8,3.9) | 1.7 (1.09, 2.66) | 0.018* |
| | Urban | 78 | 132 | 1 | | |
| Educational status | Can't read& write | 166 | 100 | 2.5 (1.4,4.8) | 0.94 (0.4, 2.18) | 0.880 |
| | Grade 1–8 | 22 | 55 | 0.6 (0.35,1.3) | 0.47 (0.12, 1.11) | 0.085 |
| | Grade 9–12 | 17 | 40 | 0.65 (0.29,1.46) | 0.82 (0.33, 2.02) | 0.666 |
| | College & above | 19 | 29 | 1 | | |
| ANC follow up | No | 100 | 34 | 4.5 (2.87,7.07) | 2.5 (1.49, 4.18) | 0.001* |
| | Yes | 124 | 190 | 1 | | |
| Gravidity | ≥5 | 143 | 88 | 5.8 (2.39,12.6) | 1.96 (0.66,5.78) | 0.225 |
| | 2–4 | 73 | 109 | 2.3 (0.97,5.25) | 1.63 (0.61,4.35) | 0.331 |
| | Primigravida | 8 | 27 | 1 | | |
| History of HDP | Yes | 5 | 10 | 0.5 (0.16,1.45) | 0.3 (0.09, 1.12) | 0.074 |
| | No | 219 | 214 | 1 | | |
| Dexamethasone given | No | 191 | 178 | 1.5 (0.91,2.44) | 1.3 (0.71, 1.06) | 0.406 |
| | Yes | 33 | 46 | 1 | | |
| Referral from another health facility | Yes | 158 | 110 | 2.5 (1.75,3.81) | 1.08 (0.61, 1.89) | 0.794 |
| | No | 66 | 114 | 1 | | |
| Delay seek care >12 hrs. | Yes | 113 | 40 | 4.7 (3.04, 7.20) | 2.5 (1.55, 4.19) | 0.000* |
| | No | 111 | 184 | 1 | | |
| Mode of delivery | Emergency C/D | 153 | 100 | 2.1 (1.05,4.20) | 1.89 (0.87,4.07) | 0.110 |
| | Instrumental delivery | 7 | 6 | 1.6 (0.45,5.69) | 0.44 (0.12, 1.65) | 0.533 |
| | SVD | 48 | 96 | 0.7 (0.33,1.40) | 0.6 (0.15, 2.69) | 0.387 |
| | Elective C/D | 16 | 22 | 1 | | |

COR: crude odds ratio, AOR: adjusted odds ratio, CI: confidence interval

*statistically significant at p-value <0.05, 1 reference value.

follow-up, and advanced maternal age were significant determinants of poor feto-maternal outcomes of antepartum hemorrhage. The absence of ANC and the delay in seeking care during pregnancy put the mother and her fetus at increased risk of antepartum hemorrhage complications. This study revealed that women who delayed seeking care for more than 12 hours were more likely to develop poor feto-maternal outcomes as compared to early health-care seekers. This is consistent with a study conducted at Mettu Karl referral hospital in Ethiopia [17]. This could be due to the absence of awareness of obstetric danger signs during pregnancy, lack of transportation, failure of the early referral system, and poor community awareness [32].

Women residing in rural areas were at increased risk of developing poor feto-maternal outcomes than those urban residents, which is similar to a study done at Jimma University's specialized hospital in Ethiopia [30]. This might be due to the distance from their home to the health facility, the absence of media in a remote area, their low socioeconomic class, or even their attitude towards ANC follow-up during pregnancy.

Women without ANC follow-up were more likely to develop poor feto-maternal outcomes as compared to their counterparts. This finding is consistent with a study conducted in Hyderabad [31]. This is because ANC is the ideal platform for important healthcare functions, including health promotion, disease prevention, screening, and early intervention. By providing appropriate evidence-based ANC, it is possible to save lives and any complications for mothers and newborns related to APH [33]. Moreover, financial constraints, the unavailability of road construction, the cost of transportation, and low community awareness about the importance of ANC affected ANC follow-up. In turn, these contribute to a delay in seeking care for early prevention of complications [34, 35].

In this study, advanced maternal age >35 years was significantly associated with poor feto-maternal outcomes of antepartum hemorrhage. This could be due to atherosclerotic alterations in the uterine blood vessels, which reduce uteroplacental blood flow and produce infarction, resulting in little placental perfusion [36–38].

There were 25,152 deliveries during the study period in all Awi Zone public hospitals. Among these, 448 had antepartum hemorrhage complicating 2.6% of them, which is in line with studies done in different countries [5, 17, 39, 40]. However, our study's finding is lower than a study done at Jimma University's specialized hospital [30]. This might be due to the difference in the level of caregiving practice in hospitals. Jimma University's specialized hospital is a tertiary hospital serving critical patients referred from different health facilities.

Placenta previa 234 (52.2%), followed by abruption placenta 190 (42.2%), were the most common causes of APH. Supporting this finding, other studies in different countries also revealed that placenta previa was the primary cause of APH [5, 41–45]. In contrast to this finding, the major cause of APH was abruption of the placenta, followed by placenta previa, according to a study done at JUSH and India [30, 31].

For 169 (75.4%) cases and 122 (54.4%) controls, their mode of delivery was a caesarian section. This is comparable with the study findings at Jimma University Hospital, three teaching hospitals in Addis Ababa, and a study conducted in India [5, 30, 45]. This increased rate of cesarean sections was due to maternal health concerns, which include hemorrhagic shock, placenta previa, and fetal distress.

The current study found 2 (0.9%) maternal deaths, which is lower than the study conducted in India and Jimma University Hospital in Ethiopia [30, 31]. This could be due to the difference in level and setup of the two hospitals. Moreover, the lifesaving care of blood transfusion was conducted for 24.1% of patients in the current study, which is higher than the number of patients transfused with blood at Jimma University Hospital.

Fifty-four (24.1%) cases were transfused with blood. This is higher than a study conducted at Jimma University Hospital in which only 18.5% of patients were transfused with at least one unit of blood. For mothers suffering from obstetric hemorrhage, blood transfusion is a life-saving element of comprehensive emergency obstetric care [30].

The current study revealed that about 78 (17.4%), 143 (63.8%), and 97 (43.3%) newborns from mothers with poor feto-maternal outcomes were stillborn, had low APGAR scores, and had low birth weights, respectively. Comparably to these findings, a study conducted at a northern Nigerian teaching hospital reported that 57.2%, 25.6%, and 42.8% of newborns had low birth weight, low Apgar score, and stillbirths, respectively [31].

Birth asphyxia (32%), and prematurity (25%), were the major neonatal complications in the current study. This is congruent with a study finding at Jimma University Hospital, three teaching hospitals in Addis Ababa, and a study done in India [5, 30, 45].

### Limitations

The limitation of this study is its retrospective design, which makes it difficult to access cards with full patient information. Therefore, future researchers will be better able to conduct follow-up studies. This study was also conducted at public hospitals, which makes its result difficult to apply to the whole population of pregnant women with antepartum hemorrhage.

## Conclusion and recommendations

This study revealed that rural residence, delay seeking care >12 hours, not having ANC follow-up, and advanced maternal age were significant determinants of poor feto-maternal outcomes of antepartum hemorrhage. The absence of ANC and the delay in seeking care during pregnancy put the mother and her fetus at increased risk of antepartum hemorrhage complications.

## Recommendations

Based on our study findings, we recommend the expansion of health education about the importance of ANC follow-up for health promotion and early detection of complications, especially for rural residents. It is also better to give attention and commitment to advocating timely marriage and pregnancy to prevent obstetric complications related to advanced age.

## Supporting information

**S1 Dataset.**
(SAV)

**S2 Dataset.**
(SPS)

## Acknowledgments

We would like to thank Assosa University for the duplication of the questionnaire.

## Author Contributions

**Conceptualization:** Ambaye Minayehu Zegeye, Fentahun Tamene Zeleke.

**Data curation:** Ambaye Minayehu Zegeye, Seid Wodajo Liben, Fentahun Tamene Zeleke, Dawit Misganaw Belay.

**Formal analysis:** Ambaye Minayehu Zegeye, Yibelu Bazezew, Wale Kumlachew, Fentahun Tamene Zeleke.

**Funding acquisition:** Ambaye Minayehu Zegeye.

**Investigation:** Ambaye Minayehu Zegeye.

**Methodology:** Ambaye Minayehu Zegeye, Fentahun Tamene Zeleke, Dawit Misganaw Belay.

**Project administration:** Ambaye Minayehu Zegeye.

**Resources:** Ambaye Minayehu Zegeye, Seid Wodajo Liben, Yaregal Dessalew Tarik, Getahun Degualeh Kebede, Yilkal Dagnaw, Fentahun Tamene Zeleke, Dawit Misganaw Belay.

**Software:** Ambaye Minayehu Zegeye, Yibelu Bazezew, Ashete Adare, Wale Kumlachew, Getahun Degualeh Kebede, Yilkal Dagnaw, Dawit Misganaw Belay.

**Supervision:** Ambaye Minayehu Zegeye, Yibelu Bazezew, Paulos Jaleta, Yaregal Dessalew Tarik, Yilkal Dagnaw.

**Validation:** Ambaye Minayehu Zegeye, Wale Kumlachew.

**Visualization:** Ambaye Minayehu Zegeye, Yibelu Bazezew, Ashete Adare, Paulos Jaleta, Wale Kumlachew, Seid Wodajo Liben, Getahun Degualeh Kebede, Fentahun Tamene Zeleke.

**Writing – original draft:** Ambaye Minayehu Zegeye.

**Writing – review & editing:** Ambaye Minayehu Zegeye, Ashete Adare, Seid Wodajo Liben, Yaregal Dessalew Tarik, Getahun Degualeh Kebede, Yilkal Dagnaw, Fentahun Tamene Zeleke, Dawit Misganaw Belay.

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
