## [Decision Letter · Decision Letter 0]

16 Aug 2022

PONE-D-22-15417DETERMINANTS OF FETO-MATERNAL OUTCOMES OF ANTEPARTUM HEMORRHAGE AMONG WOMEN WHO GAVE BIRTH IN AWI ZONE PUBLIC HOSPITALS, ETHIOPIA, 2021 .PLOS ONE

Dear Dr. Zegeye,

Thank you for submitting your manuscript to PLOS ONE. After careful consideration, we feel that it has merit but does not fully meet PLOS ONE’s publication criteria as it currently stands. Therefore, we invite you to submit a revised version of the manuscript that addresses the points raised during the review process.

Please note that we have only been able to secure a single reviewer to assess your manuscript. We are issuing a decision on your manuscript at this point to prevent further delays in the evaluation of your manuscript. Please be aware that the editor who handles your revised manuscript might find it necessary to invite additional reviewers to assess this work once the revised manuscript is submitted. However, we will aim to proceed on the basis of this single review if possible. The reviewer has suggested revisiting your statistical analyses with the aim of identifying the most important factors, and has also provided some other pointers to improve the manuscript. Please respond carefully to all of the reviewer's recommendations when preparing your revision.

We look forward to receiving your revised manuscript.

Kind regards,

Jamie Males

Editorial Office

PLOS ONE

Journal Requirements:

“We would like to thank Assosa University for financial support to this study and Debre markos University for granting permission to this study. This study was funded by the Assosa University Research and Community Service Directorate”

“NO - Include this sentence at the end of your statement: The funders had no role in study design, data collection, and analysis, decision to publish, or preparation of the manuscript.”

4. Please upload a new copy of Figure 1 as the detail is not clear. Please follow the link for more information: " ext-link-type="uri" xlink:type="simple">https://blogs.plos.org/plos/2019/06/looking-good-tips-for-creating-your-plos-figures-graphics/"
" ext-link-type="uri" xlink:type="simple">https://blogs.plos.org/plos/2019/06/looking-good-tips-for-creating-your-plos-figures-graphics/"

Reviewers' comments:

Reviewer's Responses to Questions

**Comments to the Author**

1. Is the manuscript technically sound, and do the data support the conclusions?

Reviewer #1: Yes

2. Has the statistical analysis been performed appropriately and rigorously? 

Reviewer #1: Yes

3. Have the authors made all data underlying the findings in their manuscript fully available?

Reviewer #1: No

4. Is the manuscript presented in an intelligible fashion and written in standard English?

Reviewer #1: Yes

5. Review Comments to the Author

Reviewer #1: General comments

The authors put together a good manuscript on the determinants of feto-maternal outcomes of antepartum hemorrhage among women who gave birth in Awi Zone public hospitals , Ethiopia using a case-control methodology. The authors determined that the following were the significant determinants of APH; rural residence, Delay to seek care 12 hours, not having Antenatal care follow up and advanced maternal age. However they do not provide which factors weigh more heavily than others. This also would reflect on strategies to inform policy and practice. I suggest that from their analysis the authors identify the factors in terms of statistical weight and explain the variation and reasons why some factors weigh in heavily than others

Specific comments

In the paragraph on clinical related characteristics of the cases and the controls the authors state the following- “…With regard to parity 117(52.2%) of cases were grand multipara who gave birth to five or more baby followed by multi parity 91(40.6%) and primi-para 16 (7.2%) respectively…” The authors should revise the sentence to change from the use of the word multi parity to multiparous and primiparous women for clarity of the sentence.

The following paragraph also has a list of abbreviations D C, HDp and C/D that need to be written in full for the first time to aid reviewers and readers follow and understand the article

In section 3.3 – The authors write that –“…Most of the women were attained their antenatal care follow up…”. Kindly review the sentence for clarity

6. PLOS authors have the option to publish the peer review history of their article (what does this mean?). If published, this will include your full peer review and any attached files.

Reviewer #1: **Yes: **Jackline Oluoch Aridi

---

## [Author Response · Author response to Decision Letter 0]

21 Sep 2022

we tried to address all comments and suggestions raised by reviewers.

---

## [Decision Letter · Decision Letter 1]

28 Mar 2023

PONE-D-22-15417R1DETERMINANTS OF FETO-MATERNAL OUTCOMES OF ANTEPARTUM HEMORRHAGE AMONG WOMEN WHO GAVE BIRTH IN AWI ZONE PUBLIC HOSPITALS, ETHIOPIA, 2021 .PLOS ONE

Dear Dr. Zegeye,

Thank you for submitting your manuscript to PLOS ONE. After careful consideration, we feel that it has merit but does not fully meet PLOS ONE’s publication criteria as it currently stands. Therefore, we invite you to submit a revised version of the manuscript that addresses the points raised during the review process.

ACADEMIC EDITOR:- use the current report on maternal mortality (Trends in maternal mortality)-Include the regional (SSA) perspective. What are the current estimates and the changes overtime and the major causes? - relatedly, include some statistics on Ethiopia - current estimates, causes and policies in place (see also Reviewer 3 comment). - Describe your study variables and their measurements in this study including how they were initially collected and changed during analysis. It would be better to have them in a table. -In your conclusion (abstract and main paper), include the recommendation-There are many grammar and typo issues that you need to address. - as pointed out by Reviewers 2 and 3, you need to revise your methods section- as pointed out by Reviewer 1, you need to rework on your discussion sections considering your results and the study area's context and current implementation strategies/policies, and other available evidence within or outside Ethiopia. This also applies to the recommendations. - Overall, you need to restructure your manuscript and follow the journal guideline. You can check some of the published papers for guidance.

If applicable, we recommend that you deposit your laboratory protocols in protocols.io to enhance the reproducibility of your results. Protocols.io assigns your protocol its own identifier (DOI) so that it can be cited independently in the future. For instructions see: https://journals.plos.org/plosone/s/submission-guidelines#loc-laboratory-protocols. Additionally, PLOS ONE offers an option for publishing peer-reviewed Lab Protocol articles, which describe protocols hosted on protocols.io. Read more information on sharing protocols at https://plos.org/protocols?utm_medium=editorial-emailutm_source=authorlettersutm_campaign=protocols.

We look forward to receiving your revised manuscript.

Kind regards,

Rornald Muhumuza Kananura

Academic Editor

PLOS ONE

Reviewers' comments:

Reviewer's Responses to Questions

**Comments to the Author**

1. If the authors have adequately addressed your comments raised in a previous round of review and you feel that this manuscript is now acceptable for publication, you may indicate that here to bypass the “Comments to the Author” section, enter your conflict of interest statement in the “Confidential to Editor” section, and submit your "Accept" recommendation.

Reviewer #2: All comments have been addressed

Reviewer #3: (No Response)

2. Is the manuscript technically sound, and do the data support the conclusions?

Reviewer #2: Partly

Reviewer #3: Partly

3. Has the statistical analysis been performed appropriately and rigorously? 

Reviewer #2: Yes

Reviewer #3: Yes

4. Have the authors made all data underlying the findings in their manuscript fully available?

Reviewer #2: Yes

Reviewer #3: Yes

5. Is the manuscript presented in an intelligible fashion and written in standard English?

Reviewer #2: Yes

Reviewer #3: Yes

6. Review Comments to the Author

Reviewer #2: The text still needs some refinements for publication. In the summary, it is appropriate to include only the sample that was used, 448 women, and not the total that was eligible, 651 women. Finally, consult the Medical Subject Headings (MESH) of the keywords of the study. In the inclusion method, the simple random sampling technique was used to select controls and obtain the final sample. In the results, rewrite the initial paragraph, as the construction does not make it clear whether the results of the sociodemographic characteristics are related to 651 or 448 women in the study. In figure 1, the value informed in the text for the potentially eligible population is different, in the text 651 medical records and in figure 1- 569 medical records.

Reviewer #3: The authors have conducted an interesting analysis about an important topic: DETERMINANTS OF FETO-MATERNAL OUTCOMES OF ANTEPARTUM HEMORRHAGE AMONG WOMEN WHO GAVE BIRTH IN AWI ZONE PUBLIC HOSPITALS, ETHIOPIA, 2021. I encourage them to consider the following points as they revise the manuscript.

Title: The title is clear and relevant.

Abstract

1. Background: Please add clear justification/gap why you conduct this study; there was other studies done in different areas of Ethiopia?

2. Conclusion: Please add what is your recommendation based on your findings

Introduction

1. Please better to add what figure shows about late pregnancy bleeding/APH in Ethiopia.

2. Better to add clear gap… what makes your study differ from others/why you intended to conduct this study area.

Methods

1. Needs to be clarify ‘‘…..the final calculated sample size was 448 (224 cases and 244 controls)’’.

2. Clearly rewrite the inclusion criteria….. ‘‘Inclusion Criteria Women diagnosed with APH and singleton delivery ≥weeks of gestation and who had full information needed for the study were included’’

3. Why you excluded women who diagnosed with multiple pregnancy?

4. Rewrite the operational definition of Antepartum hemorrhage it’s not correct .

Result

1. What does it mean ….651 were diagnosed and managed , and 2.6% of women get complication.

2. Please rewrite again…. ‘‘Women residing in rural areas were 70.6% times more likely to develop poor feto-maternal outcomes compared to women residing in urban areas [AOR= 1.706, 95% CI 1.09, 2.66].’’

3. How ‘‘LNMP remember’’ could be a predictor of Feto-maternal outcomes regarding APH?

Discussion

1. Needs rewriting …. Poor justification.

2. Please add the limitations of this study?

Conclusion:

1. Please better to add and correlate your recommendation with the findings

7. PLOS authors have the option to publish the peer review history of their article (what does this mean?). If published, this will include your full peer review and any attached files.

Reviewer #2: No

Reviewer #3: **Yes: **Mulualem Silesh

---

## [Author Response · Author response to Decision Letter 1]

25 May 2023

we appreciate all academic editors for their valuable review and comments.

---

## [Decision Letter · Decision Letter 2]

29 Aug 2023

PONE-D-22-15417R2DETERMINANTS OF FETO-MATERNAL OUTCOMES OF ANTEPARTUM HEMORRHAGE AMONG WOMEN WHO GAVE BIRTH IN AWI ZONE PUBLIC HOSPITALS, ETHIOPIA, 2021.PLOS ONE

Dear Dr. Zegeye,

Thank you for submitting your manuscript to PLOS ONE. After careful consideration, we feel that it has merit but does not fully meet PLOS ONE’s publication criteria as it currently stands. Therefore, we invite you to submit a revised version of the manuscript that addresses the points raised during the review process.

ACADEMIC EDITOR:

Dear authors,

This is an interesting paper that would have clinical as well as public health importance to avert avoidable deaths and morbidities due to APH. However, the paper needs to be revised.

Abstract: the background information is unnecessarily long. Avoid definitions of APH and add take home messages grounded on this study.The Introduction section mixes the problem statement with significance and rationale of the study. I would suggest restricting as 1) problem statement (that is magnitude of APH, its consequences/outcomes, etc.); 2) what is known and not in the literature (including risk factors or determinants and national efforts). In this case, the last sentence of the first paragraph should come here; 3) and then the significance and/or rationale of the study could followVariables and analysis: How distance traveled is measured? Residence could interact with other variables like delay to seek care, ANC follow-up, etc. As such, during analysis, in addition to the collinearity check, I was expecting an examination of the effect modification or interaction effect.Limitations: Any limitations regarding the missingness of data or records? And what specific variables were not measured due to the retrospective nature of the study.Discussion: Please summarize the key findings of this study in the first paragraph and then compare the findings with the literature and discuss their implications in subsequent paragraphs. For instance, paragraphs 2, 3, and 4 are not primary outcomes of this study. You might discuss them later. Discuss the key determinants first.Minor: I don’t think charts or records are source and study populations, rather mothers Please correct accordinglyRegarding health system-related variables, I feel some of them are obstetric or clinical factors like ANC follow-up and prolonged hospital stay. Think of it.Some of the recommendations are not grounded in this study.Please submit your revised manuscript by Oct 13 2023 11:59PM. If you will need more time than this to complete your revisions, please reply to this message or contact the journal office at plosone@plos.org. Please include the following items when submitting your revised manuscript:A rebuttal letter that responds to each point raised by the academic editor and reviewer(s). You should upload this letter as a separate file labeled 'Response to Reviewers'.A marked-up copy of your manuscript that highlights changes made to the original version. You should upload this as a separate file labeled 'Revised Manuscript with Track Changes'.An unmarked version of your revised paper without tracked changes. You should upload this as a separate file labeled 'Manuscript'.If applicable, we recommend that you deposit your laboratory protocols in protocols.io to enhance the reproducibility of your results. Protocols.io assigns your protocol its own identifier (DOI) so that it can be cited independently in the future. For instructions see: https://journals.plos.org/plosone/s/submission-guidelines#loc-laboratory-protocols. Additionally, PLOS ONE offers an option for publishing peer-reviewed Lab Protocol articles, which describe protocols hosted on protocols.io. Read more information on sharing protocols at https://plos.org/protocols?utm_medium=editorial-emailutm_source=authorlettersutm_campaign=protocols.

We look forward to receiving your revised manuscript.

Kind regards,

Gizachew Tadele Tiruneh, Ph.D.

Academic Editor

PLOS ONE

Journal Requirements:

Reviewers' comments:

Reviewer's Responses to Questions

**Comments to the Author**

1. If the authors have adequately addressed your comments raised in a previous round of review and you feel that this manuscript is now acceptable for publication, you may indicate that here to bypass the “Comments to the Author” section, enter your conflict of interest statement in the “Confidential to Editor” section, and submit your "Accept" recommendation.

Reviewer #3: (No Response)

2. Is the manuscript technically sound, and do the data support the conclusions?

Reviewer #3: Yes

3. Has the statistical analysis been performed appropriately and rigorously? 

Reviewer #3: Yes

4. Have the authors made all data underlying the findings in their manuscript fully available?

Reviewer #3: Yes

5. Is the manuscript presented in an intelligible fashion and written in standard English?

Reviewer #3: Yes

6. Review Comments to the Author

Reviewer #3: Thank you for your response …However, still the following points not addressed.

1. Needs to be clarify ‘‘.the final calculated sample size was 448 (224 cases and 244 controls---which yields 468)’’.

2. Why you excluded women who diagnosed with multiple pregnancy?

7. PLOS authors have the option to publish the peer review history of their article (what does this mean?). If published, this will include your full peer review and any attached files.

Reviewer #3: **Yes: **Mulualem Silesh

---

## [Author Response · Author response to Decision Letter 2]

6 Oct 2023

We have been able to incorporate changes to reflect most of the suggestions provided by the reviewers. We have highlighted (colored) the changes within the manuscript. And a point-by-point response to the reviewers’ and academic editor’s comments.

---

## [Editor Report · Decision Letter 3]

11 Jan 2024

Determinants of Feto-Maternal outcomes of antepartum hemorrhage among women who gave birth in Awi zone public hospitals, Ethiopia. A case-control study.

PONE-D-22-15417R3

Dear Dr. Zegeye,

We’re pleased to inform you that your manuscript has been judged scientifically suitable for publication and will be formally accepted for publication once it meets all outstanding technical requirements and the comments forwarded.

Kind regards,

Gizachew Tadele Tiruneh, Ph.D.

Academic Editor

PLOS ONE

Additional Editor Comments (optional):

Dear authors,

Please address the comments and language issues highlighted yellow. You might also consult Grammarly or other free language editing software.

Best,
---

## [Editor Report · Acceptance letter]

22 May 2024

PONE-D-22-15417R3 

PLOS ONE

Dear Dr. Zegeye, 

I'm pleased to inform you that your manuscript has been deemed suitable for publication in PLOS ONE. Congratulations! Your manuscript is now being handed over to our production team.

Kind regards, 

on behalf of

Dr. Gizachew Tadele Tiruneh 

Academic Editor

PLOS ONE